# Unsupervised Learning of Node Embeddings by Detecting Communities

## Abstract

We present *Deep MinCut* (DMC), an unsupervised approach to learn node embeddings for graph-structured data. It derives node representations based on their membership in communities. As such, the embeddings directly provide interesting insights into the graph structure, so that the separate node clustering step of existing methods is no longer needed. DMC learns both, node embeddings and communities, simultaneously by minimizing the *mincut loss*, which captures the number of connections between communities. Striving for high scalability, we also propose a training process for DMC based on minibatches. We provide empirical evidence that the communities learned by DMC are meaningful and that the node embeddings are competitive in different node classification benchmarks.

## 1 Introduction

Graphs are a natural representation of relations between entities in complex systems, such as social networks or information networks. To enable inference on graphs, a *graph embedding* may be learned. It comprises *node embeddings*, each being a vector-based representation of a graph's node that incorporates its relations to other nodes (Goyal & Ferrara, 2018; Hamilton et al., 2017b). While supervised node embeddings have received a lot of attention, most real-world graphs are not labelled, which calls for unsupervised learning techniques.

The main principle in unsupervised learning of node embeddings is that "similar" nodes have close embeddings in the embedding space. The similarity of nodes is often defined based on their distance in a graph, e.g., based on their co-occurrence probability in a random walk (Goyal & Ferrara, 2018; Perozzi et al., 2014; Grover & Leskovec, 2016). Recently, it was also argued that two nodes should be similar, if they are similar to a graph summary representation (Veličković et al., 2018).

In this work, we argue that node embeddings shall not only be of high quality for inference tasks, but shall also be meaningful. That is, they shall directly provide insights into interesting structures in a graph in order to avoid a potentially biased post-analysis step, e.g., through clustering of the embeddings. We therefore assess node similarity from the perspective of node communities, where the dimensions of embeddings are some unknown communities instead of some unknown latent features as in traditional techniques. Considering a community as a set of densely connected nodes with sparse connections to outside nodes, the homophily principle is restated as follows: Nodes with similar community membership characteristics shall have close embeddings. Specifically, for each node, we incorporate membership information as a probability distribution over a set of communities. Then, nodes are similar, if they are both likely and unlikely to be part of the same communities.

Since communities are generally unknown, we propose to minimize the *mincut* loss for *unsupervised* learning of communities and node embeddings *simultaneously*. Mincut loss leverages the principle that communities are well-separated if there are few connections between them (Fortunato, 2010). It is theoretically motivated as its optimal closed-form solution can be found, while its variant, the *normalized cut*, is a well-studied problem.

Aiming at a realisation of the above idea, we propose Deep MinCut (DMC), a neural network approach to minimize mincut loss. We learn node embeddings to sample one-hot vectors that represent the assignment of nodes to communities. The vectors are drawn from distributions parameterized by continuous node embeddings using Gumbel-Softmax (Jang et al., 2016; Maddison et al., 2016). This renders the process differentiable and, thus, enables joint learning of embeddings and communities.

We demonstrate the applicability of DMC in various applications. In node classification, our node embeddings turn out to outperform traditional embedding techniques (Grover & Leskovec, 2016; Perozzi et al., 2014; Veličković et al., 2018), while also revealing the graph's community structure. In community detection, by stacking mincut losses, we are able to learn a hierarchy of communities, e.g., when generating word embeddings, we can link words to topics, and topics to abstract themes.

## 2 RELATED WORK

**Graph embedding** constructs a low-dimensional model of the nodes of a graph that incorporates its structure (Hamilton et al., 2017b; Goyal & Ferrara, 2018). Embedding techniques can be classified into shallow (Perozzi et al., 2014; Grover & Leskovec, 2016) and deep approaches (Hamilton et al., 2017a; Kipf & Welling, 2016; Wu et al., 2019). Shallow approaches rely on an embedding lookup table to map nodes to embeddings. On the other hand, deep models construct a node's embedding by performing aggregation of its neighbours' embeddings.

**Unsupervised node embeddings.** While different models may be employed to embed nodes (Perozzi et al., 2014; Grover & Leskovec, 2016), the majority of *unsupervised* learning techniques leverage a contrastive loss function, such as skipgram loss (Hamilton et al., 2017b) or infomax loss (Veličković et al., 2017). The encoder is trained such that, given a scoring function, a high score is given to positive samples, whereas negative samples receive a low score. For skipgram loss, the positive samples are nodes that are close in a random walk, while the negative samples are randomly selected from other graph nodes. A drawback of the random-walk objective is that it can only capture local information around a node (Perozzi et al., 2014; Grover & Leskovec, 2016).

For infomax loss, positive samples are nodes in the original graph, whereas negative samples are nodes in randomly corrupted graphs. While infomax loss can capture the global structure, its performance is highly dependent on the corruption strategy (Veličković et al., 2018). Since embeddings also need to be learned for the corrupted graphs, it further suffers from high training time. While our proposed method also considers the global graph structure, it differs in that our approach is non-constrastive, i.e., it does not require unnecessary learning of negative samples. Moreover, due to the nature of our loss function, we are able to learn the node embeddings as well as their clusters.

**Community detection** is a well-studied problem with many applications (Fortunato, 2010; Fortunato & Hric, 2016). While numerous techniques for community detection have been proposed, we focus on those that generate node embeddings, such as spectral methods (Newman, 2006b;a; White & Smyth, 2005). Spectral methods operate either on the modularity matrix (Newman, 2006b) or the Laplacian matrix (White & Smyth, 2005). While embeddings may be learned by reconstructing these matrices (Wang et al., 2016), existing methods leverage matrix factorization, which does not scale to large graphs. Closest to our work is (Nazi et al., 2019), which proposes a partition loss function for graph partitioning. As the work focuses on graph partitioning, the loss function aims for balanced partitions based on the number of nodes and, therefore, is not applicable in our setting.

## 3 EMBEDDINGS AND COMMUNITY DETECTION

**Graphs.** We consider a directed, weighted graph $\mathcal{G} = \{\mathbb{V}, \mathbb{F}\}$ with nodes $\mathbb{V} = \{v_i\}$ and edges $\mathbb{F} = \{(v_i, v_j) \mid v_i \in \mathbb{V} \land v_j \in \mathbb{V}\}$, each edge $(v_i, v_j)$ being assigned a weight $s_{(v_i, v_j)} \in \mathbb{R}$. Such a graph can also be represented by its adjacency matrix $\boldsymbol{A}$ of size $n \times n$, where each row and column represents a node in $\mathcal{G}$ and a cell $\mathbf{A}_{ij}$ denotes the edge weight. Note that we allow self-loops in the graph, but not multi-edges between nodes. Also, edge weights can be negative. We also consider attributed graphs in which nodes have features. We denote the node features matrix as $\mathbf{F} \in \mathbb{R}^{n \times D}$.

**Communities.** We denote by $\mathbb{C} = \{\mathbb{C}_1, \mathbb{C}_2, \cdots, \mathbb{C}_k\}$ the set of $k$ disjoint communities of graph $\mathcal{G}$, where $\bigcup_{i=1}^{k} \mathbb{C}_i = \mathbb{V}$ and $\forall\, \mathbb{C}_i \neq \mathbb{C}_j, \mathbb{C}_i \cap \mathbb{C}_j = \emptyset$. The assignment of nodes to communities is captured by a *membership matrix* $\mathbf{P} \in \{0, 1\}^{n \times k}$ with rows representing nodes in $\mathcal{G}$ and columns representing communities in $\mathbb{C}$. As each node is only assigned to one community, the rows of $\mathbf{P}$ are one-hot vectors, where $\mathbf{P}_{ij} = 1$ if node $v_i$ is assigned to community $\mathbb{C}_j$.

Assuming the membership matrix $\mathbf{P}$ is already known, the number of cross-connections between communities $\mathbb{C}_i$ and $\mathbb{C}_j$ can be captured by the non-diagonal elements of the adjacency matrix $\mathbf{C}$ of

the *quotient graph*, where the nodes are communities:

$$\mathbf{C} = \mathbf{P}^T \mathbf{A} \mathbf{P} \tag{1}$$

On the other hand, elements $\mathbf{C}_{ii}$ capture the number of connections within community $\mathbb{C}_i$.

**Mincut loss.** While community detection is a well-studied problem, there is no consensus on the precise notion of a community (Fortunato, 2010). A common principle is that well-separated communities have more connections inside than across communities. Hence, communities are detected by minimizing the number of connections between them, as captured by the following loss function:

$$\mathcal{L}_{\mathbf{P}}(\mathbf{A}) = -\sum_i^k \mathbf{C}_{ii} = -Tr(\mathbf{P}^T \mathbf{A} \mathbf{P}) \tag{2}$$

where $Tr(\mathbf{X})$ is the trace of matrix $\mathbf{X}$. We call the loss function in Equation 2 *mincut loss*, as it aims to minimize the number of connections between communities.

**Degenerated cases.** Minimizing Equation 2 may lead to degenerated cases, where all nodes are assigned to one community while the others are empty (Fortunato, 2010). In practice, there are two solutions to this problem. If there is prior knowledge on the communities (e.g., they shall have equal size), a respective constraint is added to the mincut loss. Another approach is to minimize the *normalized cut* (Shi & Malik, 2000; Zhang & Rohe, 2018), which is defined as follows:

$$ncut(\mathbb{C}) = \sum_{i=1}^k \frac{cut(\mathbb{C}_i, \mathbb{C}_{-i})}{assoc(\mathbb{C}_i, \mathbb{C}_{-i})} = \sum_{i=1}^k \frac{\mathbf{P}_{:,i}^T(\mathbf{D} - \mathbf{A})\mathbf{P}_{:,i}}{\mathbf{P}_{:,i}^T \mathbf{D} \mathbf{P}_{:,i}} = Tr(\frac{\mathbf{P}^T(\mathbf{D} - \mathbf{A})\mathbf{P}}{\mathbf{P}^T \mathbf{D} \mathbf{P}}) = Tr(\frac{\mathbf{P}^T \mathbf{L} \mathbf{P}}{\mathbf{P}^T \mathbf{D} \mathbf{P}})$$

where $\mathbb{C}_{-i}$ denotes the set of communities except $\mathbb{C}_i$ and $assoc(\mathbb{C}_i, \mathbb{C}_{-i})$ is the total degree of nodes in community $\mathbb{C}_i$. Then, the normalized cut (*normcut*) loss can be captured as follows:

$$\mathcal{L}_{\mathbf{P}}(\mathbf{A}) = Tr(\frac{\mathbf{P}^T \mathbf{L}^{sym} \mathbf{P}}{\mathbf{P}^T \mathbf{P}}) \tag{3}$$

where $\mathbf{L}^{sym} = \mathbf{I} - \mathbf{D}^{-1/2} \mathbf{A} \mathbf{D}^{1/2}$ is the symmetric Laplacian matrix with $\mathbf{D}$ be the degree matrix of $\mathbf{A}$ and and $\mathbf{L} = \mathbf{D} - \mathbf{A}$ is the Laplacian matrix.

**Graph-like data.** For ease of presentation, mincut loss is formulated based on graphs. Yet, it can be applied to any problem comprising a set $\mathbb{T}$ of items and a kernel function $k : \mathbb{T} \times \mathbb{T} \to \mathbb{R}$ that assigns weights to item pairs. This creates a kernel matrix $\mathbf{K}$ that can be considered as the adjacency matrix of the items. Mincut loss is designed to separate items into subsets such that the connection strength between every pair of subsets is minimized, which also means the coherence of each subset is maximized. Hence, mincut loss is applicable to a wide range of *unsupervised* problems.

## 4 DEEP MINCUT

We first discuss the spectral approach to find optimal solutions for normcut and mincut loss, as it provides a baseline technique for comparison. We then introduce Deep MinCut along with an efficient training process based on minibatches.

### 4.1 SPECTRAL APPROACH

Since $\mathbf{P}$ is a binary matrix, optimizing the normcut loss to find $\mathbf{P}$ is an NP-hard problem (Fortunato, 2010). Following traditional approaches in community detection (Fortunato, 2010; White & Smyth, 2005), we relax $\mathbf{P}$ from a binary matrix to a real matrix $\mathbf{H} \in \mathbb{R}^{n \times k}$. However, for the relaxed matrix $\mathbf{H}$ to be meaningful, it needs to retain the following semantic constraint from matrix $\mathbf{P}$ that each node belongs to only one community: $\widetilde{\mathbf{h}}_i \widetilde{\mathbf{h}}_j^T = 0$ where $\widetilde{\mathbf{h}}_i$ is the $i$-th *column* of the matrix $\mathbf{H}$. The matrix $\mathbf{H}$ that minimizes Equation 3 can be found by eigendecomposition of the adjacency matrix $\mathbf{L}^{sym}$. This is captured by the following theorem.

**Theorem 1.** *Let $\mathbf{L}^{sym}$ be the normalized Laplacian matrix of a graph $\mathcal{G}$ of size $n$ and its eigende-composition $\mathbf{L}^{sym} = \mathbf{Q} \mathbf{\Lambda} \mathbf{Q}^T$. We also denote $\lambda_1, \cdots, \lambda_k$ to be $k$ smallest eigenvalues of $\mathbf{L}^{sym}$ and*

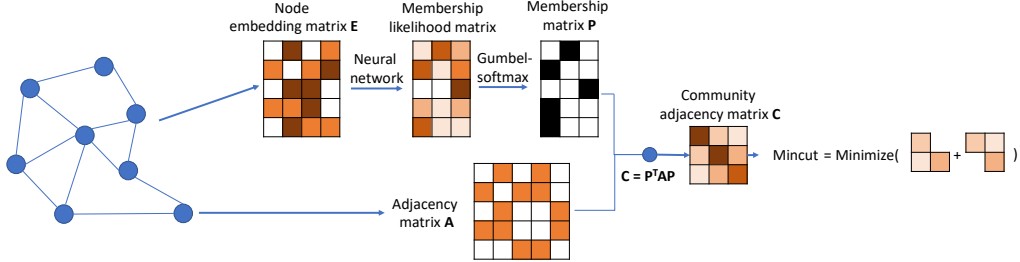

Figure 1: Learning embeddings by detecting communities.

*their respective eigenvectors $\mathbf{q}_1, \cdots, \mathbf{q}_k$. Let $\mathbf{H} \in \mathbb{R}^{n \times k}$ be a matrix that satisfies $\widetilde{\mathbf{h}}_i \widetilde{\mathbf{h}}_j^T = 0$. Then, $\mathcal{L}_{\mathbf{H}}(\mathbf{L}^{sym}) = Tr(\frac{\mathbf{H}^T \mathbf{L}^{sym} \mathbf{H}}{\mathbf{H}^T \mathbf{H}})$ is minimized when the $i$-th column vector of $\mathbf{H}$ is parallel with the $i$-th eigenvector, i.e., $\widetilde{\mathbf{h}}_i \uparrow\uparrow \mathbf{q}_i$.*

It is worth noting that closed-form solutions for the mincut loss can be found in a similar manner where the matrix $\mathbf{H}$ that minimizes the mincut loss can be constructed from the *largest* eigenvectors of the adjacency matrix $\mathbf{A}$. The proofs of these theorems can be found in Appendix C.

## 4.2 DEEP MINCUT - A NEURAL NETWORK APPROACH

Although analytical solutions to Equation 2-3 can be found by eigendecomposition, such approach to minimizing the normcut loss is infeasible for large graphs. We therefore propose a neural network approach called Deep MinCut to learn node embeddings and communities at the same time. Our framework is illustrated in Figure 1, which we explain in detail in the remainder.

**Learning the membership matrix.** To detect communities, we want to learn the membership matrix $\mathbf{P}$ that minimizes the normalized cut $\mathcal{L}_{\mathbf{P}}(\mathbf{A}) = Tr(\frac{\mathbf{P}^T \mathbf{L}^{sym} \mathbf{P}}{\mathbf{P}^T \mathbf{P}})$. Recall that $\mathbf{P}$ captures the assignment of nodes to communities. Intuitively, this assignment is based on a node's role in the graph and the graph structure, which is also the information that shall be encoded in a node embedding (Grover & Leskovec, 2016; Hamilton et al., 2017a). Hence, we propose to compute the membership matrix $\mathbf{P}$ based on node embeddings.

An embedding matrix $\mathbf{H}$, which contains all the node embeddings, is derived by an encoder $\mathcal{E}_\theta : \mathbb{V} \to \mathbb{R}^d$. The encoder $\mathcal{E}_\theta$ encodes every node in $\mathbb{V}$ to a $d$-dimensional space, where $\theta$ denote parameters. Any existing node embedding techniques such as graph convolutional encoders (Hamilton et al., 2017b; Wu et al., 2019) or shallow encoders (Grover & Leskovec, 2016; Perozzi et al., 2014) can be used to realize $\mathcal{E}_\theta$, since the parameters $\theta$ can be learned during optimization of the normalized cut. The node embeddings in $\mathbf{H}$ can also be used for downstream tasks such as node classification or link prediction.

**Differential sampling.** To obtain $\mathbf{P}$ from $\mathbf{H}$, we sample a one-hot vector $\boldsymbol{p}$ of $\mathbf{P}$ from the corresponding node embedding $\boldsymbol{h}$ of $\mathbf{H}$. Intuitively, $\mathbf{H}_{ij}$ shall capture the likelihood that the $i$-th node is assigned to the $j$-th community. As such, we consider the elements $h_1, \ldots, h_k$ of $\boldsymbol{h}$ to be the unnormalized class probabilities of a $k$-dimensional categorical distribution. Then, by sampling from this distribution, we are able to obtain the one-hot vector $\boldsymbol{p}$. Sampling from this distribution can be done using the Gumbel-max trick (Gu et al., 2018; Niu et al., 2019) where the non-zero element of the one-hot vector $\boldsymbol{p} = (\mathrm{p}_1, \ldots, \mathrm{p}_k)$ is found as follows:

$$\mathrm{p}_i = \begin{cases} 1, & \text{if } i = \arg\max_j (\mathrm{h}_j + g) \\ 0, & \text{otherwise} \end{cases}$$

where $g \sim Gumbel(0, 1)$ is a sample from the standard Gumbel distribution.

Note that the sampling process is still non-differentiable as the $\arg\max$ operation is discontinuous since the Gumbel-max trick only makes sampling from a categorical distribution an optimization problem. By replacing the $\arg\max$ function with the differentiable *softmax*, however, we render

---

**Algorithm 1:** Computation of normcut loss.

---

**input** : Adjacency matrix $\mathbf{A}$, $\tau$, straight-through $st$, embedding matrix $\mathbf{H}$
**output:** Normcut loss $\mathcal{L}$

1 $\mathbf{P} = gumbel\_softmax(\mathbf{H}, \tau, st)$ ;                          // Sample one-hot vectors from node embeddings
2 $\mathbf{C} = \mathbf{H}^T \mathbf{A} \mathbf{H}$ ;                          // Adjacency matrix of the quotient graph.
3 $\boldsymbol{q} = \mathbf{1} \mathbf{C}$ ;                          // Compute the association of communities. $\mathbf{1}$ is the vector of ones.
4 $\boldsymbol{d} = diagonal(\mathbf{C})$ ;                          // Diagonal vector of $\mathbf{C}$.
5 $\boldsymbol{l} = (\boldsymbol{q} - \boldsymbol{d})/\boldsymbol{q}$ ;                          // Compute the normcut loss
6 $\mathcal{L} = \sum \boldsymbol{l}$ ;
7 **return** $\mathcal{L}$ ;

---

the whole process differentiable (Maddison et al., 2016; Jang et al., 2016). That is, an element $\mathrm{p}_i$ of $\mathbf{p}$ can be sampled from the corresponding row $\mathbf{h}$ of $\mathbf{H}$ as follows:

$$\mathrm{p}_i = \frac{\exp(\mathrm{x}_i/\tau)}{\sum_i^k \exp(\mathrm{x}_i/\tau)}$$

where $\mathrm{x}_i = \mathrm{h}_i + g$ and $\tau$ is a temperature hyperparameter. Gumbel-Softmax (GS) not only allows to sample discrete values from a continuous distribution, but it is also differentiable. The latter is important for learning the parameters of $\mathcal{E}_\theta$ using backpropagation. As the temperature $\tau \to 0$, the row $\mathbf{p}$ approaches the one-hot vector. The whole computation process of the normcut loss from an adjacency matrix and the node embeddings is shown in Algorithm 1.

**Straight-through Gumbel-Softmax.** Since we want $\mathbf{p}$ to be close to a one-hot vector, we need to set $\tau$ to be close to 0. However, a small temperature leads to a high variance of gradients which makes the training process slow to converge (Jang et al., 2016). To this end, we use the Straight-through Gumbel-Softmax which allows us to set a high temperature while obtaining one-hot vector for $\mathbf{p}$. This is done by taking the $\arg \max$ of $\mathbf{p}_i$ to construct the one-hot vector in the forward pass, while in the backward pass, $\mathbf{p}_i$ is used to compute the gradients. The trade-off is that there is a mismatch between the forward and backward pass which makes Straight-through GS a biased estimator. However, it performs well in practice (Choi et al., 2018; Gu et al., 2018; Niu et al., 2019).

**Hierarchical community detection.** Several mincut losses may be stacked to learn a hierarchy of items. For instance, another mincut loss can be applied to the adjacency matrix $\mathbf{C}$ to learn super-communities. Then, communities and super-communities can be learned in an end-to-end manner, similar to the hierarchical pooling framework in (Ying et al., 2018). However, our approach is unsupervised and may thus be used in applications where labels are not available.

### 4.3 MINIBATCH TRAINING

**Sampling method.** To improve scalability of DMC, parameters shall be learned with batches of nodes, instead of the whole adjacency matrix. This is equivalent to approximating mincut loss with sampled subgraphs. However, a simple random sampling of nodes to construct a subgraph is not sufficient as the subgraph may not be connected. Hence, optimization of normcut loss is non-trivial.

Against this background, we propose to construct an ego-network for each node in the graph and use this ego-network as the subgraph. This sampling procedure is similar to the neighbourhood sampling method proposed by Hamilton et al. (2017a). Given a node $v \in \mathbb{V}$, its ego-network of depth $d$ is the induced subgraph obtained from a sample of all nodes with a distance of at most $d$ to $v$. Sampling is done at each level, with replacement of a fixed amount of neighbours. This is to make the subgraphs to have equal size. To create a batch of size $b$, we create $b$ such ego-networks.

**Theoretical motivation.** We provide a theoretical motivation on why it is possible to approximate the normcut loss function with sampled subgraphs. Let $\mathbf{K} \in \mathbb{R}^{m \times k}$ be the embedding matrix of the subgraph $\mathcal{S}$ of size $m$. We also denote the adjacency matrix of this subgraph as $\widetilde{\mathbf{A}}$, its degree matrix as $\widetilde{\mathbf{D}}$, and its Laplacian matrix as $\widetilde{\mathbf{L}}$. The following theorem shows that we can approximate normcut loss to a certain degree with high probability using the subgraph of size $m < n$ where $n$ is the number of nodes in the original graph.

**Theorem 2.** *Let* $\mathcal{L} = \sum_{i=1}^k \frac{\mathbf{H}_{:,i}^T \mathbf{L} \mathbf{H}_{:,i}}{\mathbf{H}_{:,i}^T \mathbf{D} \mathbf{H}_{:,i}}$, $\widetilde{\mathcal{L}} = \sum_{i=1}^k \frac{\mathbf{K}_{:,i}^T \widetilde{\mathbf{L}} \mathbf{K}_{:,i}}{\mathbf{K}_{:,i}^T \widetilde{\mathbf{D}} \mathbf{K}_{:,i}}$ *be the normcut loss of the graph* $\mathcal{G}$ *and subgraph* $\mathcal{S}$ *with adjacency matrices* $\mathbf{A}, \widetilde{\mathbf{A}}$ *respectively. Let* $a, b \in \mathbb{R}$ *be the upper and lower*

Table 1: Statistics of the datasets

| Dataset | Nodes | Edges | Features | Classes |
|---|---|---|---|---|
| **Cora** (Sen et al., 2008) | 2,708 | 5,429 | 1,433 | 7 |
| **Citeseer** (Sen et al., 2008) | 3,327 | 4,732 | 3,703 | 6 |
| **Pubmed** (Namata et al., 2012) | 19,717 | 44,338 | 500 | 3 |
| **Wiki** (Grover & Leskovec, 2016) | 4,777 | 184,812 | n/a | 40 |

*bound of $\mathbf{A}$ then if $\mathbf{H}$ is a binary matrix and elements of $\mathbf{A}$ and $\widetilde{\mathbf{A}}$ are i.i.d then given an $\varepsilon \geq 0$,*
$P(|\frac{\widetilde{\mathcal{L}}-\mathcal{L}}{\mathcal{L}}| \leq \varepsilon) \geq 1 - 2k(\exp(\frac{-(n\varepsilon)^2}{128k^2(b-a)^2}) + \exp(\frac{(-m\varepsilon)^2}{128k^2(b-a)^2}) + \exp(\frac{-n^2\varepsilon^2}{64k(b-a)^2}) + \exp(\frac{-m^2\varepsilon^2}{64k(b-a)^2}) +$
$2\exp(\frac{-m\varepsilon^2}{394k}) + 2\exp(\frac{-n}{8k}))$.

Theorem 2 (proof in Appendix C) states that we can choose a batch size such that the difference in the approximated loss and the true loss is small with high probability. Other factors that affect this probability are the bounds $a, b$ of its adjacency matrix and the embedding size $k$. Finally, the probability also depends on how accurately we approximate the loss function, as controlled by $\varepsilon$.

## 5 EXPERIMENTS

### 5.1 NODE EMBEDDING EXPERIMENTS

**Setup.** We evaluate the quality of our embeddings for node classification on four datasets, see Table 1. Cora, Citeseer and Pubmed are paper citation networks where a label represents the domain of a paper. Wiki is a word adjacency graph with the word labels being their POS tags.

In this experiment, we use a one-layer Graph Convolutional Network (Kipf & Welling, 2016) as the encoder for DMC and also report the results obtained with the spectral approach. We compare DMC with several baselines. First, we compare against community detection techniques that generate node embeddings, such as DANMF (Ye et al., 2018), M-NMF (Wang et al., 2017) and GAP (Nazi et al., 2019). Second, we include unsupervised node embedding methods that use contrastive loss, such as DeepWalk (Perozzi et al., 2014) and DGI (Veličković et al., 2018). For all methods that involve randomization, we train three models with different seeds. The node embedding size is consistently set to 128. The obtained node embeddings are used to learn a logistic classifier. Instead of a fixed train/test split, we use 50 random splits and report the mean accuracy and standard deviation as suggested by Shchur et al. (2018).

**Results.** Table 2 highlights the benefits of generating embeddings by community detection for node classification. Our technique outperforms the baseline methods in three out of four benchmarks. The largest gap is observed for the Wiki dataset, which can be explained by the non-attributed nature of this dataset. DMC considers the whole structure of the graph, whereas most baseline techniques consider only the neighbourhood surrounding a node. While DGI is able to incorporate the whole graph, it relies more on node features, which is less beneficial for non-attributed graphs. The spectral approach underperforms significantly on the bibliographic datasets as it only uses the structure information in the graph. In addition, the optimal solution to normcut loss may not be the best embeddings for node classification, as the node labels are more correlated to node features.

In addition to node embeddings, DMC also learns how to cluster the embeddings. We compare the cluster quality obtained by our approach with the best baseline for node classification, which is DGI. For DGI, we use k-means to cluster the node embeddings into several clusters where the number of clusters is the number of classes. Then, we compare the cluster quality obtained using DMC and DGI on two metrics: Normalized Mutual Information (NMI) and Homogeneity (HG).

Table 3 shows that the cluster quality obtained by DMC is significantly higher than the one by DGI. For instance, the NMI scores with DMC are $7\times$ better than those with DGI on the Cora dataset. This illustrates the benefits of jointly learning node embeddings and their clusters. This is particularly important in unsupervised settings, where the learned embeddings are fed into a downstream task, such as node classification or graph analysis through clustering.

Table 2: Node classification results.

|  | Method | Cora | Citeseer | Pubmed | Wiki |
|---|---|---|---|---|---|
| Community detection | GAP | $0.768\pm1.1e^{-2}$ | $0.663\pm4.8e^{-3}$ | $0.754\pm5.9e^{-3}$ | $0.596\pm4.1e^{-3}$ |
|  | M-NMF | $0.775\pm0.9e^{-3}$ | $0.562\pm0.9e^{-3}$ | $0.763\pm0.3e^{-3}$ | $0.64\pm1.3e^{-2}$ |
|  | NMF | $0.514\pm1.4e^{-2}$ | $0.443\pm9.3e^{-3}$ | $0.672\pm7.4e^{-3}$ | $0.485\pm4.9e^{-3}$ |
|  | Feat.+Comm. | $0.749\pm0.8e^{-3}$ | $0.709\pm0.9e^{-3}$ | $\mathbf{0.851}\pm0.2e^{-3}$ | $0.659\pm0.1e^{-2}$ |
| Contrastive loss | DeepWalk | $0.670\pm2.6e^{-3}$ | $0.479\pm3.3e^{-3}$ | $0.722\pm4.9e^{-3}$ | $0.491\pm2.6e^{-3}$ |
|  | DGI | $0.813\pm0.3e^{-3}$ | $0.711\pm7.4e^{-3}$ | $0.835\pm3.6e^{-3}$ | $0.568\pm1.2e^{-3}$ |
| Normcut loss | **DMC** (Ours) | $\mathbf{0.839}\pm1.8e^{-3}$ | $\mathbf{0.713}\pm4.1e^{-3}$ | $0.833\pm2.2e^{-3}$ | $\mathbf{0.659}\pm4.5e^{-3}$ |
|  | Spectral | 0.303 | 0.206 | 0.397 | 0.581 |

| Dataset | Metric | DGI | DMC |
|---|---|---|---|
| Cora | NMI | 0.061 | **0.429** |
|  | HG | 0.068 | **0.475** |
| Citeseer | NMI | 0.059 | **0.212** |
|  | HG | 0.067 | **0.264** |
| Pubmed | NMI | 0.172 | **0.215** |
|  | HG | 0.191 | **0.277** |
| Wiki | NMI | 0.272 | **0.286** |
|  | HG | **0.345** | 0.339 |

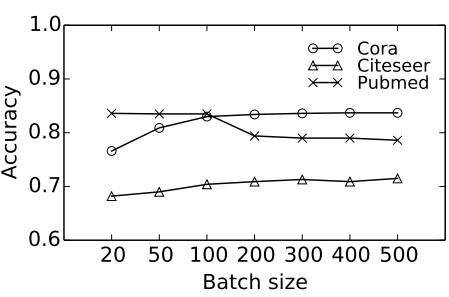

Table 3: DMC vs. DGI on cluster quality

Figure 2: Effect of batch size

## 5.2 COMMUNITY DETECTION EVALUATION

**Setup.** Next, we aim to show the applicability of normcut loss to a graph-like setting, such as learning word embeddings. Here, the nodes are words and the connection weight between two words is measured by the following "kernel" function:

$$k(w_i, w_j) = max(\frac{\log(\#(w_i, w_j))}{\log(\#w_i)\log(\#w_j)}, 0)$$

where $\#w_i$ is the number of times word $w_i$ appears in the corpus, while $\#(w_i, w_j)$ is the number of times the words appear together. The adjacency matrix obtained using the above function is the PPMI matrix, a well-established concept in NLP (Levy & Goldberg, 2014). Following Yin & Shen (2018), we construct a word corpus of 10000 words that appear >100 times in the Text8 corpus (Mahoney, 2011). Words are said to appear together if they are within a window of five.

We further analyse the quality of the communities constructed by DMC. As those are represented by the dimensions of the node embeddings (in this case, word embeddings), high-quality communities correspond to explainable word topics. We evaluate explainability by a word intrusion test. We create a set of five words for each dimension by selecting the top-4 words and a single low-ranked word. Human workers on MTurk are asked to detect one word per dimension that does not belong to the respective set. The detection precision then measures explainability. We compare DMC against explainable word embeddings techniques, OIWE (Luo et al., 2015), Sparse Coding (SC) (Faruqui et al., 2015), and Non-Negative Sparse Coding (NNSC) (Faruqui et al., 2015). For a qualitative analysis, we also report the words with the largest embedding values along exemplary dimensions.

**Results.** Table 4 shows that DMC outperforms state-of-the-art methods in precision of the word intrusion test. Note that NNSC, SC, and OIWE require additional data such as existing word embeddings as input, whereas our methods learn explainable word embeddings directly on a word corpus. Table 5 shows that the top-ranked words indeed assign a meaning to each dimension (here, the first three dimensions concern medieval literature, DC comics, and transportation). By stacking two normcut losses, DMC is also able to learn a hierarchy of words and topics. Figure 4 gives an example, where super-topic #18 captures IT-related words, while supertopic #3 is related to religion.

Table 4: Test Precision

|  | **Precision** |
| --- | --- |
| NNSC | 35.85% |
| SC | 47% |
| OIWE | 91.01% |
| DMC | **95.24**% |

Table 5: Top-5 words for the first five dimensions

| Dim #1 | Dim #2 | Dim #3 | Dim #4 | Dim #5 |
| --- | --- | --- | --- | --- |
| medieval | created | flight | full | internet |
| earliest | features | pilot | job | network |
| scholars | dc | navy | fair | client |
| renaissance | adaptation | passenger | offered | server |
| classical | batman | aviation | calling | servers |

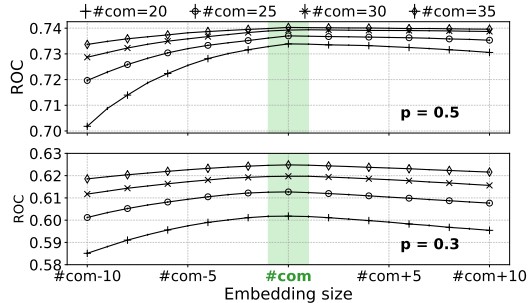

Figure 3: Embedding size vs. #communities

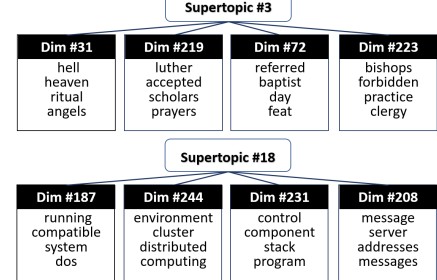

Figure 4: Hierarchy of words/topics

### 5.3 EFFECTS OF MINIBATCH TRAINING

**Setup.** We evaluate the effects of minibatch training on the classification accuracy on three citation networks by varying the batch size from 20 to 500.

**Results.** Confirming our theoretical analysis, Figure 2 shows that the difference between the approximated loss and the true loss depends on the batch size. With increasing batch sizes, the accuracy on the Cora and Citeseer datasets increases as well. However, after an initial sharp increase, the differences become smaller. This shows the robustness of our approach to the batch size. Moreover, we observe a reversed trend on the Pubmed dataset, which is the largest among the citation graphs. We believe that this can be attributed to the stochasticity of minibatch training, which renders outlier nodes to be less important as the subgraphs can only cover parts of the whole graph.

### 5.4 RELATION BETWEEN EMBEDDING SIZE AND NUMBER OF COMMUNITIES

**Setup.** The aim of this experiment is two-fold. First, we aim to show the merit of the mincut loss. While the normcut prevents degenerate cases, the mincut loss is useful if we have prior knowledge about the graph structure. Second, we want to analyze the effect of the embedding size w.r.t the number of communities. For this experiment, we need a ground truth number of communities, which is why we rely on the Stochastic Block Model (SBM), a well-established benchmark for community detection (Chen et al., 2019; Fortunato & Hric, 2016; Fortunato, 2010). Using SBM, a graph with a known number of communities ($\#com$) is generated. Then, we construct node embeddings with varying dimensionality using the spectral approach for mincut loss with an additional balancing constraint on the community sizes. The parameters of SBM are discussed in detail in Appendix B. Node embeddings are assessed for link prediction with the ROC metric (avg over 100 runs). We chose two values for parameter $p$, the probability of an edge between two nodes in a community.

**Results.** Figure 3 confirms that *"there is a sweet spot for the dimensionality, [..] neither too small, nor too large"* (Arora et al., 2016). Our results provide one possible explanation for the dimensionality trade-off. If the embedding size is smaller than the number of communities, unrelated communities are combined, lowering the ROC. If the embedding size is larger than the number of communities, communities are split up further. This also decreases the ROC, but not as drastically, since the model has higher capacity. In practice, the embedding quality tends to increase and then stabilize, with increasing embedding sizes, due to the communities often having different sizes.

## 6 Conclusion

We presented a novel perspective on unsupervised learning of node embeddings. Following the idea of community detection, we proposed Deep MinCut (DMC), an approach to minimize the mincut loss function to learn node embeddings and communities simultaneously. DMC learns node embeddings that are not only of high quality, but are also meaningful as they capture the graph's structure. We demonstrated the value of node embeddings learned with mincut loss in diverse experiments.

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

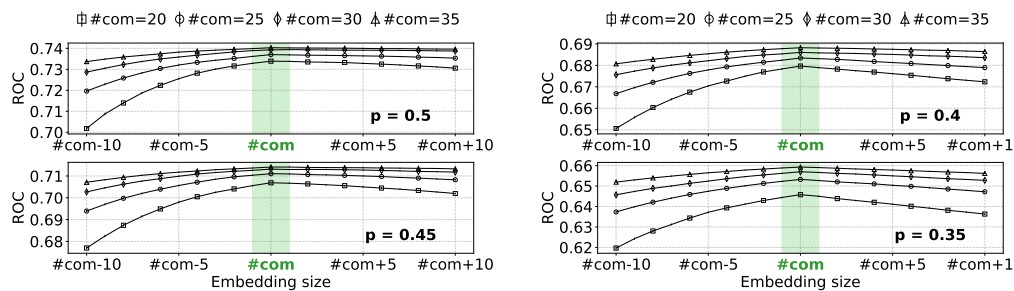

Figure 5: Relationship between embedding size and #communities

## A    ADDITIONAL EXPERIMENTAL RESULTS

### A.1    RELATIONSHIP BETWEEN EMBEDDING SIZE AND NUMBER OF COMMUNITIES

Taking up the discussion in Section 5.4, Figure 5 provides additional results with different values of $p$ for the Stochastic Block Model. We observe that the ROC scores increase with $p$. This is expected as larger $p$ values correlate with a clearer community structure, which yields better link prediction. On the other hand, with smaller $p$ values, we observe the dimensionality trade-off better. As $p$ decreases, the graphs converge towards random graphs with no community structure. As a result, it is easier for the model to cluster nodes into communities incorrectly. This effect becomes more severe when the gap between the embedding size and the number of communities is larger. The reason being that the model is forced to cluster nodes into communities that may be larger or smaller than actual communities.

## B    DETAILS ON DATASETS AND HYPERPARAMETERS

### B.1    HYPERPARAMETERS

The embedding sizes of all methods are set to 128 to achieve a fair comparison. For methods that use deep models (DMC, DGI, GAP), we use a learning rate of 0.001 and Adam as the optimizer. All methods are trained for a fixed number of epochs, namely 3000. For DMC, since we use Straight-Through Gumbel-Softmax, we fix the temperature to be 1. For minibatch DMC, we use a batch size of 100 and two-layer neighbourhood sampling, where 10 and 25 neighbours are used in the first and second layer. For word embedding experiment, we use a shallow encoder for DMC. For experiments with k-means, we take the average scores (NMI and HG) over 10 runs.

### B.2    DATASETS

**Node embedding experiments.** Cora (Sen et al., 2008), Citeseer (Sen et al., 2008) and Pubmed (Namata et al., 2012) are three standard benchmark datasets for node classification. Traditionally, a fixed train/test split is used to evaluate the node embeddings. However, Shchur et al. (2018) showed that using random splits for evaluation is fairer for different methods. As these datasets are attributed, we also include the non-attributed Wiki dataset Grover & Leskovec (2016). For this dataset, we initialize the node features using truncated SVD, setting the feature size to 128.

**Word embedding experiments.** To learn the embeddings, we use the Text8 corpus Mahoney (2011) which is a standard large-scale benchmark for various NLP tasks. To construct the PPMI matrix, we leverage a library that is publicly available[1] and use the same setting as in Yin & Shen (2018) to build the vocabulary and count the co-occurrences of word pairs: A window size of 5 and a minimum occurrence of a word of 100.

---

[1]https://github.com/ziyin-dl/word-embedding-dimensionality-selection

**Stochastic Block Model (SBM).** To build graphs from the SBM, we leverage the networkx library.[2] The parameters of SBM include the number of communities $\#ncom$, community sizes and the probability of edges between nodes in a community $p$. We use the balanced SBM where the community sizes are set to be equal to 100, while we vary $p$ from $0.3$ to $0.5$ to generate the graphs. From $p$, we can compute the probability of connection across communities as $\bar{p} = \frac{1-p}{1-\#ncom}$. When $p \sim \bar{p}$, the graph becomes a random graph with no community structure.

### B.3 Hardware

Experiments were conducted on a workstation with an Intel Core i7-6700K CPU @ 4.00GHz with 32 GB RAM and an Nvidia GTX 1080Ti GPU with 12 GB and a server with 2 nodes. Each node includes an AMD Ryzen 1900X CPU with 64GB RAM and 1 Nvidia GTX 1080Ti GPU with 12 GB.

## C Proofs

### C.1 Analytical solution for mincut loss

**Theorem 1.** *Let $\mathbf{M}$ be a positive semi-definite matrix of size $n \times n$ and its eigendecomposition $\mathbf{M} = \mathbf{Q}\mathbf{\Lambda}\mathbf{Q}^T$. We also denote $\lambda_1, \cdots, \lambda_m$ to be $m$ largest eigenvalues of $\mathbf{M}$ and their respective eigenvectors $\mathbf{q}_1, \cdots, \mathbf{q}_m$. Let $\mathbf{H} \in \mathbb{R}^{n \times m}$ be a matrix that satisfies $\langle \widetilde{\mathbf{h}}_i, \widetilde{\mathbf{h}}_j \rangle = 0$ then $\mathcal{L}_\mathbf{H}(\mathbf{M}) = -Tr(\mathbf{H}^T\mathbf{M}\mathbf{H})$ is minimized when the $i$-th column vector of $\mathbf{H}$ is parallel with the $i$-th eigenvector i.e. $\overline{\mathbf{h}}_i \upuparrows \mathbf{q}_i$.*

*Proof.* Note that we can factorize any matrix $\mathbf{H} = \mathbf{U}\mathbf{X}$ by a unitary matrix $\mathbf{U}$ of size $n \times m$ and a diagonal matrix $\mathbf{X}$ of size $m$. We denote $\mathbf{X}\mathbf{X}^T = diag(x_1, x_2, ..., x_m)$ and $\mathbf{U}_{1:k}$ is the matrix of $k$ leading columns of $\mathbf{U}$. We also denote $\mathbf{I}_{1:k}$ to be the identity matrix of size $k \times k$. Then, we have:

$$\mathcal{L}_\mathbf{H}(\mathbf{M}) = -Tr(\mathbf{H}^T\mathbf{M}\mathbf{H}) = -Tr(\mathbf{H}\mathbf{H}^T\mathbf{M}) = -Tr(\mathbf{U}\mathbf{X}\mathbf{X}^T\mathbf{U}^T\mathbf{M})$$

$$= -Tr(\sum_1^m (\mathbf{U}_{1:k}(x_k - x_{k+1})\mathbf{I}_{1:k}\mathbf{U}_{1:k}^T\mathbf{M})) = -\sum_1^m (x_k - x_{k+1})Tr(\mathbf{U}_{1:k}^T\mathbf{M}\mathbf{U}_{1:k})$$

By applying the Poincare Separation Theorem Abadir & Magnus (2005), we have $\forall k = \overline{1,m}, Tr(\mathbf{U}_{1:k}^T\mathbf{M}\mathbf{U}_{1:k})$ is maximized iff $\forall k = \overline{1,m}$, the column vectors of $\mathbf{U}_{1:k}$ are proportional to $k$ leading eigenvectors of $\mathbf{M}$. In addition, as the column vectors of $\mathbf{U}$ or $\mathbf{U}_{1:k}$ are proportional to the column vectors of $\mathbf{H}$, $\mathcal{L}_\mathbf{H}(\mathbf{M})$ is minimized iff the column vectors of $\mathbf{H}$ are proportional to $m$ leading eigenvectors of $\mathbf{M}$. In other words, the column vectors of $\mathbf{H}$ are parallel with $m$ leading eigenvectors of $\mathbf{M}$. □

### C.2 Analytical solution for normcut loss

**Theorem 2.** *Let $\mathbf{L}^{sym}$ be the normalized Laplacian matrix of a graph $\mathcal{G}$ of size $n$ and its eigendecomposition $\mathbf{L}^{sym} = \mathbf{Q}\mathbf{\Lambda}\mathbf{Q}^T$. We also denote $\lambda_1, \cdots, \lambda_k$ to be $k$ smallest eigenvalues of $\mathbf{L}^{sym}$ and their respective eigenvectors $\mathbf{q}_1, \cdots, \mathbf{q}_k$. Let $\mathbf{H} \in \mathbb{R}^{n \times k}$ be a matrix that satisfies $\widetilde{\mathbf{h}}_i\widetilde{\mathbf{h}}_j^T = 0$ then $\mathcal{L}_\mathbf{H}(\mathbf{L}^{sym}) = Tr(\frac{\mathbf{H}^T\mathbf{L}^{sym}\mathbf{H}}{\mathbf{H}^T\mathbf{H}})$ is minimized when the $i$-th column vector of $\mathbf{H}$ is parallel with the $i$-th eigenvector i.e. $\widetilde{\mathbf{h}}_i \upuparrows \mathbf{q}_i$.*

*Proof.* We can see that if we multiply $\mathbf{h}_i$ with an arbitrary number, the value of $\mathcal{L}_\mathbf{H}(\mathbf{L}^{sym})$ is unchanged. So that we can assume $\mathbf{H}^T\mathbf{H} = \mathbf{I}$. With this assumption, the problem becomes finding $\mathbf{H}$ to minimize $Tr(\mathbf{H}^T\mathbf{L}^{sym}\mathbf{H})$.

Let $\mu_1 \geq \mu_2 \geq \cdots \geq \mu_k$ be the eigenvalues of $\mathbf{H}^T\mathbf{L}^{sym}\mathbf{H}$ and $\lambda_1 \geq \lambda_2 \geq \cdots \geq \lambda_n$ be the eigenvalues of $\mathbf{L}^{sym}$. Then, according to Poincare'seperation theorem, we have:

---

[2]https://networkx.github.io/

$$\sum_{i=1}^{k} \lambda_{n-k+i} \leq \sum_{i=1}^{k} \mu_i \leq \sum_{i=1}^{k} \lambda_i \tag{4}$$

Noting that $\sum_{k=1}^{r} \mu_k = \text{Tr}(\mathbf{H}^T \mathbf{L}^{sym} \mathbf{H})$. This means $\text{Tr}(\mathbf{H}^T \mathbf{L}^{sym} \mathbf{H})$ is minimized at $\sum_{i=1}^{k} \lambda_{n-k+i}$ when $\mathbf{h}_i$ is proportion with $k$ smallest eigenvectors of $\mathbf{L}^{sym}$. $\qquad \square$

While the closed-form solutions for the mincut and normcut loss can both be constructed from eigendecomposition, there are difference in application. The normcut loss is able to prevent degenerated cases since they do not correspond to the optimal loss value. On the other hand, for these degenerated cases, the mincut loss is minimal.

## C.3 APPROXIMATED NORMCUT LOSS

Before proving Theorem 2, we provide the following lemmas.

**Lemma 1.** *Given* $x = \frac{a}{b}, y = \frac{c}{d}$. *If* $P\left(\left|\frac{a-c}{c}\right| \leq \varepsilon\right) \geq 1 - p$ *and* $P\left(\left|\frac{b-d}{d}\right| \leq \varepsilon\right) \geq 1 - q$ *with small* $\varepsilon$ *then* $P\left(\left|\frac{x-y}{y}\right| \leq 2\varepsilon\right) \geq 1 - (p+q)$

*Proof.* We consider the case where $x, y \geq 0$ as similar result can be proved for $x, y < 0$. Let $a = (1+\alpha)c$ and $b = (1+\beta)d$. Then, we can rewrite the above statements as

$$P(|\beta| \leq \varepsilon) \geq 1 - p$$

$$P(|\alpha| \leq \varepsilon) \geq 1 - q$$

This also means:

$$\Rightarrow P(|\beta| \leq \varepsilon, |\alpha| \leq \varepsilon) \geq 1 - (p+q)$$

Moreover, when $|\beta| \leq \varepsilon$ and $|\alpha| \leq \varepsilon$ with small $\varepsilon$, we have

$$x = \frac{(1+\alpha)c}{(1+\beta)d} \Rightarrow x \geq \frac{(1-\varepsilon)c}{(1+\varepsilon)d} \geq \frac{(1-\varepsilon)}{(1+\varepsilon)} y \geq (1-2\varepsilon)y$$

Similarly,

$$x \leq \frac{(1+\varepsilon)c}{(1-\varepsilon)d} \leq (1+2\varepsilon)y$$

So that, if $|\beta| \leq \varepsilon$ and $|\alpha| \leq \varepsilon$, and a small value of $\varepsilon$, we have

$$\left|\frac{x-y}{y}\right| \leq 2\varepsilon$$

Therefore, we have $P\left(\left|\frac{x-y}{y}\right| \leq 2\varepsilon\right) \geq 1 - (p+q)$ with small $\varepsilon$ $\qquad \square$

**Lemma 2.** *Let* $\mathrm{x}_1, \mathrm{x}_2, ..., \mathrm{x}_m$ *be* $m$ *independent random variables such that* $P(\mathrm{x}_i = 1) = b_i$. *Let* $X = \sum_{i=0}^{n} \mathrm{x}_i$ *and* $\mu = \mathbb{E}[X]$. *Then, for* $0 < \delta < 1$, $P(X \leq (1-\delta)\mu) \leq \exp(-\mu\delta^2/2)$

**Corollary 1.** *Let* $\mathbf{H} \in \{0,1\}^{n \times k}$ *be an arbitrary binary matrix where the number of* $1$ *elements in* $\mathbf{H}$ *is* $n$. *Let* $p_i$ *be the ratio of* $1$ *in column* $\widetilde{\mathbf{h}}_i$, *then,* $P(p_i \geq \frac{1}{2k}) \geq 1 - \exp(\frac{-n}{8k})$

*Proof.* Applying Lemma 2 with $\mathrm{x}_i$ is $p_i$ $\forall i = \overline{1, k}$, $\mu = \mathbb{E}[p_i] = \frac{1}{k}$ and $\delta = 1/2$. $\qquad \square$

**Lemma 3.** *(Chernoff bound) Let* $x_1, x_2, ..., x_m$ *is* $m$ *random variables with Poisson distribution with* $P(x_j = 1) = p_i, X = \sum_{j=1}^{m} x_j$ *and,* $\mu = \mathbb{E}[X]$. *We have,* $\forall \delta \in (0, 1)$:

$$P\left(\left|\frac{X-\mu}{\mu}\right| \geq \delta\right) \leq 2exp(-\mu\frac{\delta^2}{3})$$

**Lemma 4.** *(Hoeffding's bound) Let $x_1, x_2, ..., x_m$ is $m$ independent random variables. $\mathbb{E}[x_i] = \mu$, $P(a \leq Y_i \leq b) = 1 \forall i$. With arbitrary reals $a$ and $b$, We have,*

$$P\left(\left|\frac{1}{n}\sum_{i=1}^{n} x_i - \mu\right| \leq \delta\right) \geq 1 - 2exp\left(\frac{-2n\delta^2}{(b-a)^2}\right)$$

**Theorem 3.** *Let $\mathcal{G}$ be a graph with its adjacency matrix $\mathbf{A}$ and its subgraph $\mathcal{S}$ with adjacency matrix $\widetilde{\mathbf{A}}$. Let $a, b \in \mathbb{R}$ be the upper and lower bound of $\mathbf{A}$ then if $\mathbf{H} \in \{0,1\}^{n \times k}$ and $\mathbf{K} \in \{0,1\}^{m \times k}$ and elements of $\mathbf{A}$ and $\widetilde{\mathbf{A}}$ are i.i.d then given an $\varepsilon \geq 0$, $P(|\frac{\widetilde{\boldsymbol{k}}_i^\top \widetilde{\mathbf{A}} \widetilde{\boldsymbol{k}}_i - c.\widetilde{\boldsymbol{h}}_i^\top \mathbf{A} \widetilde{\boldsymbol{h}}_i}{c.\widetilde{\boldsymbol{h}}_i^\top \mathbf{A} \widetilde{\boldsymbol{h}}_i}| \leq \frac{\varepsilon}{2}) \geq 1 - 2(\exp(\frac{-(n\varepsilon)^2}{128k^2(b-a)^2}) + \exp(\frac{(-m\varepsilon)^2}{128k^2(b-a)^2}) + \exp(\frac{-m\varepsilon^2}{394k}) + \exp(\frac{-n}{8k})) \ \forall i = \overline{1, n}$ where $\widetilde{\boldsymbol{h}}_i, \widetilde{\boldsymbol{k}}_i$ are the $i$-th column vectors of $\mathbf{H}, \mathbf{K}$ and $c = \frac{m^2}{n^2}$.*

*Proof.* Let $n(\mathbf{x})$ be the counting function for the number of 1 elements in a binary vector or matrix. We also denote $p_i$ to be the ratio of 1 elements in vector $\widetilde{\boldsymbol{h}}_i$. As $\mathbf{K}$ is sampled uniformly from $\mathbf{H}$ and with $n$ large enough, we can assume that each element of $\widetilde{\boldsymbol{k}}_i$ is equally chosen with probability $p_i$.

Then, $n(\widetilde{\boldsymbol{k}}_i)$ is a random variable for the number of 1 elements in vector $\widetilde{\boldsymbol{k}}_i$. We also have $n(\widetilde{\boldsymbol{k}}_i) = \sum_{j=1}^{m} \mathbf{K}_{j,i}$ which means $\mathbb{E}[n(\widetilde{\boldsymbol{k}}_i)] = mp_i$ with $P(\mathbf{K}_{j,i} = 1) = p_i$.

Then, by applying bound (Lemma 3 with $X = n(\widetilde{\boldsymbol{k}}_i)$, $\delta = \frac{\varepsilon}{8}$ and $\mu = \mathbb{E}[n(\widetilde{\boldsymbol{k}}_i)] = mp_i$, we have

$$P\left(\left|\frac{n(\widetilde{\boldsymbol{k}}_i) - mp_i}{mp_i}\right| \geq \frac{\varepsilon}{8}\right) \leq 2\exp\left(-mp_i\frac{\varepsilon^2}{192}\right)$$

Denote $c = \frac{m^2}{n^2}$, we have $n(\widetilde{\boldsymbol{h}}_i) = np_i$ which means $mp_i = \sqrt{c}.n(\widetilde{\boldsymbol{h}}_i)$. Therefore, we have:

$$P\left(\left|\frac{n(\widetilde{\boldsymbol{k}}_i) - \sqrt{c}n(\widetilde{\boldsymbol{h}}_i)}{\sqrt{c}n(\widetilde{\boldsymbol{h}}_i)}\right| \leq \frac{\varepsilon}{8}\right) \geq 1 - 2\exp\left(-mp_i\frac{\varepsilon^2}{192}\right)$$

$$\Leftrightarrow P\left(\left|\frac{n(\widetilde{\boldsymbol{k}}_i)^2 - cn(\widetilde{\boldsymbol{h}}_i)^2}{cn(\widetilde{\boldsymbol{h}}_i)^2}\right| \leq \frac{\varepsilon}{4}\right) \geq 1 - 2\exp\left(-mp_i\frac{\varepsilon^2}{192}\right) \tag{5}$$

Note that $n(\widetilde{\boldsymbol{k}}_i \widetilde{\boldsymbol{k}}_i^\top) = n(\widetilde{\boldsymbol{k}}_i)^2$ and $n(\widetilde{\boldsymbol{h}}_i \widetilde{\boldsymbol{h}}_i^\top) = n(\widetilde{\boldsymbol{h}}_i)^2$ which make the above inequality equivalent with:

$$P\left(\left|\frac{n(\widetilde{\boldsymbol{k}}_i \widetilde{\boldsymbol{k}}_i^\top) - cn(\widetilde{\boldsymbol{h}}_i \widetilde{\boldsymbol{h}}_i^\top)}{cn(\widetilde{\boldsymbol{h}}_i \widetilde{\boldsymbol{h}}_i^\top)}\right| \leq \frac{\varepsilon}{4}\right) \geq 1 - 2\exp\left(-mp_i\frac{\varepsilon^2}{192}\right) \tag{6}$$

Since elements of $\mathbf{A}$ are i.i.d and bounded by $a, b$, we have $\widetilde{\boldsymbol{h}}_i^\top \mathbf{A} \widetilde{\boldsymbol{h}}_i = \sum_{j \in \mathbb{J}} Y_j$ with $\mathbb{J} \subseteq [1, n^2]$ and $|\mathbb{J}| = n(\widetilde{\boldsymbol{h}}_i)$. Similarly, $\widetilde{\boldsymbol{k}}_i^\top \widetilde{\mathbf{A}} \widetilde{\boldsymbol{k}}_i = \sum_{j \in \mathbb{T}} Y_j$ with $\mathbb{T} \subseteq \mathbb{J}$ and $|\mathbb{T}| = n(\widetilde{\boldsymbol{k}}_i)$.

Applying lemma 4, with $\widetilde{\boldsymbol{h}}_i^\top A \widetilde{\boldsymbol{h}}_i$, $\widetilde{\boldsymbol{k}}_i^\top B \widetilde{\boldsymbol{k}}_i$ and $\delta = \frac{\varepsilon}{8}$, we have

$$P\left(\left|\widetilde{\boldsymbol{h}}_i^\top \mathbf{A} \widetilde{\boldsymbol{h}}_i - n(\widetilde{\boldsymbol{h}}_i \widetilde{\boldsymbol{h}}_i^\top)\mu\right| \leq \frac{n(\widetilde{\boldsymbol{h}}_i \widetilde{\boldsymbol{h}}_i^\top)\varepsilon}{8}\right) \geq 1 - 2\exp\left(\frac{-n(\widetilde{\boldsymbol{h}}_i \widetilde{\boldsymbol{h}}_i^\top)\varepsilon^2}{32(b-a)^2}\right)$$

and,

$$P\left(\left|\widetilde{\boldsymbol{k}}_i^\top \widetilde{\mathbf{A}} \widetilde{\boldsymbol{k}}_i - n(\widetilde{\boldsymbol{k}}_i \widetilde{\boldsymbol{k}}_i^\top)\mu\right| \leq \frac{n(\widetilde{\boldsymbol{k}}_i \widetilde{\boldsymbol{k}}_i^\top)\varepsilon}{8}\right) \geq 1 - 2\exp\left(\frac{-m(\widetilde{\boldsymbol{k}}_i \widetilde{\boldsymbol{k}}_i^\top)\varepsilon^2}{32(b-a)^2}\right)$$

Moreover, since $n(\widetilde{\boldsymbol{k}}_i \widetilde{\boldsymbol{k}}_i^\top) = (m.p_i)^2$ and $n(\widetilde{\boldsymbol{h}}_i \widetilde{\boldsymbol{h}}_i^\top) = (n.p_i)^2$, we have

$$P\left(\left|\frac{\widetilde{\boldsymbol{k}}_i^\top \widetilde{\mathbf{A}} \widetilde{\boldsymbol{k}}_i - n(\widetilde{\boldsymbol{k}}_i \widetilde{\boldsymbol{k}}_i^\top)\mu}{n(\widetilde{\boldsymbol{k}}_i \widetilde{\boldsymbol{k}}_i^\top)}\right| \leq \frac{\varepsilon}{8}, \left|\frac{\widetilde{\boldsymbol{h}}_i^\top \mathbf{A} \widetilde{\boldsymbol{h}}_i - n(\widetilde{\boldsymbol{h}}_i \widetilde{\boldsymbol{h}}_i^\top)\mu}{n(\widetilde{\boldsymbol{h}}_i \widetilde{\boldsymbol{h}}_i^\top)}\right| \leq \frac{\varepsilon}{8}\right)$$

$$\geq 1 - 2\left(\exp\left(\frac{-(np_i\varepsilon)^2}{32(b-a)^2}\right) + \exp\left(\frac{(-mp_i\varepsilon)^2}{32(b-a)^2}\right)\right) \tag{7}$$

$$\Rightarrow P\left(\frac{\left|\widetilde{\boldsymbol{k}}_i^\top \widetilde{\mathbf{A}}\widetilde{\boldsymbol{k}}_i - \frac{n(\widetilde{\boldsymbol{k}}_i\widetilde{\boldsymbol{k}}_i^\top)}{n(\widetilde{\boldsymbol{h}}_i\widetilde{\boldsymbol{h}}_i^\top)}\widetilde{\boldsymbol{h}}_i^\top \mathbf{A}\widetilde{\boldsymbol{h}}_i\right|}{\frac{n(\widetilde{\boldsymbol{k}}_i\widetilde{\boldsymbol{k}}_i^\top)}{n(\widetilde{\boldsymbol{h}}_i\widetilde{\boldsymbol{h}}_i^\top)}\widetilde{\boldsymbol{h}}_i^\top A\widetilde{\boldsymbol{h}}_i} \leq \frac{\varepsilon}{4}\right)$$

$$\geq 1 - 2(\exp(\frac{-(np_i\varepsilon)^2}{32(b-a)^2}) + \exp(\frac{(-mp_i\varepsilon)^2}{32(b-a)^2})) \quad (8)$$

Applying Lemma 1 for inequality 6 and 8 , we have,

$$P(|\frac{\widetilde{\boldsymbol{k}}_i^\top \widetilde{\mathbf{A}}\widetilde{\boldsymbol{k}}_i - c.\widetilde{\boldsymbol{h}}_i^\top \mathbf{A}\widetilde{\boldsymbol{h}}_i}{c.\widetilde{\boldsymbol{h}}_i^\top \mathbf{A}\widetilde{\boldsymbol{h}}_i}| \leq \frac{\varepsilon}{2}) \geq 1 - 2(\exp(\frac{-(np_i\varepsilon)^2}{32(b-a)^2})$$

$$+ \exp(\frac{(-mp_i\varepsilon)^2}{32(b-a)^2}) + \exp(\frac{-mp_i\varepsilon^2}{192})) \quad (9)$$

From Corollary 1, we also have

$$P(p_i \geq \frac{1}{2k}) \geq 1 - \exp(\frac{-n}{8k}) \quad (10)$$

From inequality (9) and (10) , we have,

$$P\left(\left|\frac{\widetilde{\boldsymbol{k}}_i^\top \widetilde{\mathbf{A}}\widetilde{\boldsymbol{k}}_i - c.\widetilde{\boldsymbol{h}}_i^\top \mathbf{A}\widetilde{\boldsymbol{h}}_i}{c.\widetilde{\boldsymbol{h}}_i^\top \mathbf{A}\widetilde{\boldsymbol{h}}_i}\right| \leq \frac{\varepsilon}{2}\right) \geq 1 - 2(\exp(\frac{-(n\varepsilon)^2}{128k^2(b-a)^2})$$

$$+ \exp(\frac{(-m\varepsilon)^2}{128k^2(b-a)^2}) + \exp(\frac{-m\varepsilon^2}{394k}) + \exp(\frac{-n}{8k})) \quad (11)$$

$\square$

**Theorem 4.** *Let $\mathcal{G}$ be a graph with its adjacency matrix $\mathbf{A}$ and its subgraph $\mathcal{S}$ with adjacency matrix $\widetilde{\mathbf{A}}$. Let $a, b \in \mathbb{R}$ be the upper and lower bound of $\mathbf{A}$ then if $\mathbf{H} \in \{0,1\}^{n \times k}$ and $\mathbf{K} \in \{0,1\}^{m \times k}$ and elements of $\mathbf{A}$ and $\widetilde{\mathbf{A}}$ are i.i.d then given an $\varepsilon \geq 0$, $P\left(\left|\frac{\widetilde{\boldsymbol{k}}_i^\top \widetilde{\mathbf{A}}1 - c.\widetilde{\boldsymbol{h}}_i^\top \mathbf{A}1}{c.\widetilde{\boldsymbol{h}}_i^\top \mathbf{A}1}\right| \leq \frac{\varepsilon}{2}\right) \geq 1 - 2(\exp(\frac{-n^2\varepsilon^2}{64k(b-a)^2}) + \exp(\frac{-m^2\varepsilon^2}{64k(b-a)^2}) + \exp(\frac{-m\varepsilon^2}{394k})) + \exp(\frac{-n}{8k})) \; \forall i = \overline{1, n}$ and $c = \frac{m^2}{n^2}$.*

*Proof.* (Proof sketch) Theorem 4 can be proven in the same manner as Theorem 3. $\square$

Given the results in Theorem 4 and Theorem 3, we can now prove Theorem 2.

**Theorem 5.** *Let $\mathcal{L} = \sum_{i=1}^k \frac{\widetilde{\boldsymbol{h}}_i^T \mathbf{L}\widetilde{\boldsymbol{h}}_i}{\widetilde{\boldsymbol{h}}_i^T \mathbf{D}\widetilde{\boldsymbol{h}}_i}, \widetilde{\mathcal{L}} = \sum_{i=1}^k \frac{\widetilde{\boldsymbol{k}}_i^T \widetilde{\mathbf{L}}\widetilde{\boldsymbol{k}}_i}{\widetilde{\boldsymbol{k}}_i^T \widetilde{\mathbf{D}}\widetilde{\boldsymbol{k}}_i}$ be the normcut loss of the graph $\mathcal{G}$ and subgraph $\mathcal{S}$ with adjacency matrices $\mathbf{A}, \widetilde{\mathbf{A}}$ respectively. Let $a, b \in \mathbb{R}$ be the upper and lower bound of $\mathbf{A}$ then if $\mathbf{H}$ is a binary matrix and elements of $\mathbf{A}$ and $\widetilde{\mathbf{A}}$ are i.i.d then given an $\varepsilon \geq 0$, $P(|\frac{\widetilde{\mathcal{L}}-\mathcal{L}}{\mathcal{L}}| \leq \varepsilon) \geq 1 - 2k(\exp(\frac{-(n\varepsilon)^2}{128k^2(b-a)^2}) + \exp(\frac{(-m\varepsilon)^2}{128k^2(b-a)^2}) + \exp(\frac{-n^2\varepsilon^2}{64k(b-a)^2}) + \exp(\frac{-m^2\varepsilon^2}{64k(b-a)^2}) + 2\exp(\frac{-m\varepsilon^2}{394k}) + 2\exp(\frac{-n}{8k}))$*

*Proof.* First, since $\mathbf{L} = \mathbf{D} - \mathbf{A}$, $\mathcal{L} = \sum_{i=1}^k \frac{\widetilde{\boldsymbol{h}}_i^T \mathbf{D}\widetilde{\boldsymbol{h}}_i}{\widetilde{\boldsymbol{h}}_i^T \mathbf{D}\widetilde{\boldsymbol{h}}_i} - \sum_{i=1}^k \frac{\widetilde{\boldsymbol{h}}_i^T \mathbf{A}\widetilde{\boldsymbol{h}}_i}{\widetilde{\boldsymbol{h}}_i^T \mathbf{D}\widetilde{\boldsymbol{h}}_i} = k - \sum_{i=1}^k \frac{\widetilde{\boldsymbol{h}}_i^T (\mathbf{A})\widetilde{\boldsymbol{h}}_i}{\widetilde{\boldsymbol{h}}_i^T \mathbf{D}\widetilde{\boldsymbol{h}}_i}$. This means optimizing $\mathcal{L}, \widetilde{\mathcal{L}}$ is equivalent to optimizing $\mathcal{L} = \sum_{i=1}^k \frac{\widetilde{\boldsymbol{h}}_i^T \mathbf{A}\widetilde{\boldsymbol{h}}_i}{\widetilde{\boldsymbol{h}}_i^T \mathbf{D}\widetilde{\boldsymbol{h}}_i}, \widetilde{\mathcal{L}} = \sum_{i=1}^k \frac{\widetilde{\boldsymbol{k}}_i^T \widetilde{\mathbf{A}}\widetilde{\boldsymbol{k}}_i}{\widetilde{\boldsymbol{k}}_i^T \widetilde{\mathbf{D}}\widetilde{\boldsymbol{k}}_i}$.

Second, observe that both $\mathcal{L}$ and $\widetilde{\mathcal{L}}$ are the sum of d terms. Thus, proving $\left|\frac{\mathcal{L}-\widetilde{\mathcal{L}}}{\mathcal{L}}\right| \leq \varepsilon$ is similar to prove

$$\left|\frac{\mathcal{L}_i - \widetilde{\mathcal{L}}_i}{\mathcal{L}_i}\right| \leq \varepsilon, \forall i = \overline{1, k} \quad (12)$$

in which $\mathcal{L}_i = \frac{\widetilde{\boldsymbol{h}}_i^\top \mathbf{A}\widetilde{\boldsymbol{h}}_i}{\widetilde{\boldsymbol{h}}_i^\top \mathbf{D}\widetilde{\boldsymbol{h}}_i}, \widetilde{\mathcal{L}}_i == \frac{\widetilde{\boldsymbol{k}}_i^\top \widetilde{\mathbf{A}}\widetilde{\boldsymbol{k}}_i}{\widetilde{\boldsymbol{k}}_i^\top \widetilde{\mathbf{D}}\widetilde{\boldsymbol{k}}_i}$.

---

**Algorithm 2:** Minibatch forward propagation of Deep MinCut

---

**input** : Adjacency matrix $\mathbf{A}$ of graph $\mathcal{G} = \{\mathbb{V}, \mathbb{F}\}$
       Temperature $\tau$
       Straight-through $st$
       Encoder $\mathcal{E}_\theta$
       Node feature matrix $\mathbf{F}$
       Sampling method $f$
**output:** Node embedding matrix $\mathbf{H}$
       Community assignment $\mathbf{P}$

  // 1. Sampling minibatches of nodes
1  $\mathbb{U} = f(\mathbb{V})$
2  $\mathbf{A}' = \mathbf{A}[\mathbb{U}, \mathbb{U}]$
3  $\mathbf{F}' = \mathbf{F}[\mathbb{U}]$
  // 2. Compute the node embeddings
4  $\mathbf{H} = \mathcal{E}_\theta(\mathbf{A}', \mathbf{F}')$ ;                // Embed nodes in batch using the encoder
  // 3. Compute the mincut loss
5  $\mathbf{P} = gumbel\_softmax(\mathbf{H}, \tau, st)$ ;      // Sample one-hot vectors from node embeddings
6  $\mathbf{C} = \mathbf{H}^T \mathbf{A}' \mathbf{H}$ ;               // Adjacency matrix of the quotient graph.
7  $\boldsymbol{q} = \mathbf{1}\mathbf{C}$ ;       // Compute the association of communities. $\mathbf{1}$ is the vector of ones.
8  $\boldsymbol{d} = diagonal(\mathbf{C})$ ;                 // Diagonal vector of $\mathbf{C}$.
9  $\boldsymbol{l} = (\boldsymbol{q} - \boldsymbol{d})/\boldsymbol{q}$ ;               // Compute the normcut loss
10 $\mathcal{L} = \sum \boldsymbol{l}$

---

From Lemma 1, to prove inequality 12, we need to have the following results:

$$\left| \frac{\widetilde{\boldsymbol{k}}_i^\top \widetilde{\mathbf{A}} \widetilde{\boldsymbol{k}}_i - c.\widetilde{\boldsymbol{h}}_i^\top \mathbf{A} \widetilde{\boldsymbol{h}}_i}{c.\widetilde{\boldsymbol{h}}_i^\top \mathbf{A} \widetilde{\boldsymbol{h}}_i} \right| \leq \frac{\varepsilon}{2} \tag{13}$$

And, the second is

$$\left| \frac{\widetilde{\boldsymbol{k}}_i^\top \widetilde{\mathbf{D}} \widetilde{\boldsymbol{k}}_i - c.\widetilde{\boldsymbol{h}}_i^\top \mathbf{D} \widetilde{\boldsymbol{k}}_i}{c.\widetilde{\boldsymbol{h}}_i^\top \mathbf{D} \widetilde{\boldsymbol{k}}_i} \right| \leq \frac{\varepsilon}{2} \tag{14}$$

$$\Leftrightarrow \left| \frac{\widetilde{\boldsymbol{k}}_i^\top \widetilde{\mathbf{A}} \mathbf{1} - c.\widetilde{\boldsymbol{h}}_i^\top \mathbf{A} \mathbf{1}}{c.\widetilde{\boldsymbol{h}}_i^\top \mathbf{A} \mathbf{1}} \right| \leq \frac{\varepsilon}{2}, \forall i = \overline{1, k} \tag{15}$$

where $c = \frac{m^2}{n^2}$. Inequality 13 is proven in Theorem 3 while Inequality 15 is proven in Theorem 4.

$\square$

# D   Algorithm

Algorithm 2 illustrates one iteration in the forward pass of DMC. The forward pass include 3 steps. In the first step, we sample a subset of nodes of the graph (Line 1) using a sampling strategy. In our work, we use a neighbourhood sampling strategy as discussed in Section 4.3. From the sampled nodes, we extract a submatrix from the adjacency matrix $\mathbf{A}$ in Line 2 and from the node feature matrix $\mathbf{F}$ in Line 3. The second step involves constructing the node embeddings for the sampled nodes. We use an encoder $\mathcal{E}_\theta$ to embed the nodes in Line 3. The encoder can be shallow which does not use the node features or a deep encoder. In our node classification experiment, we use 1-layer GCN as discussed in Section 5.1 and a shallow encoder for our word embedding experiment. In the last step, we use the node embeddings to compute the mincut loss as shown in Figure 1. From the loss, we can backpropagate and update the parameters $\theta$ of the encoder using any automatic differentation framework.

