# OpenReview forum: "Unsupervised Learning of Node Embeddings by Detecting Communities"
_ICLR.cc/2020/Conference — Reject_

### Official Review · AnonReviewer2 · 2019-10-22
**Official Blind Review #3**

**Rating:** 3

**Review:**

This paper is reporting an unsupervised approach to learn node embeddings and communities simultaneously by minimizing the mincut loss function. This approach programs the data through encoder to generate membership likehood matrix H, and then generates corresponding membership matrix P using node selection. The matrix P and adjacency matrix are coupled to generate community adjacency matrix to minimize the cutting. However, despite such attractive points, the novelty and strength of this study is not outstanding enough for publication in ICLR. Details comments are as follows,
1. Similar work such as the methods of transferring matrix E to H then to P have already been published , which reduce the novelty of this study.
2. Though there is a schematic, the learning embedding process is still not described clearly. More details of the algorithms need to be discussed to such as how to get the node embedding matrix E.

**Experience Assessment:**

I have published one or two papers in this area.

**Review Assessment: Checking Correctness Of Derivations And Theory:**

I assessed the sensibility of the derivations and theory.

**Review Assessment: Checking Correctness Of Experiments:**

I assessed the sensibility of the experiments.

**Review Assessment: Thoroughness In Paper Reading:**

I read the paper at least twice and used my best judgement in assessing the paper.

---

> ### Author Response · Authors · 2019-11-12
> **Response to Reviewer 2**
>
> 1. We are not aware of any similar works on generating matrix P from the embedding matrix E that is also capable of supporting end-to-end learning. It would be great if the reviewer could provide us some pointers so that we could compare.
> 2. We also provide a summary of the paper in response to Reviewer 1. Regarding the process to get the node embedding matrix E, we use different types of encoders. For node classification experiment, we use 1-layer GCN as the encoder. For word embedding experiments, we use shallow encoders. These are mentioned in Section 5.1 and 5.2.

---

### Official Review · AnonReviewer3 · 2019-10-23
**Official Blind Review #3**

**Rating:** 3

**Review:**

Although this paper seems to only combine existing techniques in community detection and node embedding into a co-train process. The idea is simple and easy understood and the paper is well-written. Theoretical analysis is provided for the approximation error for the sampling strategy. However, major concerns are:

1. Experimental results show that co-training node embedding and community detection can improve the performance for node classification. The improvements may result from the assumption that papers with the same class label are associated with the same community in the citation graph. However, in the dataset, there are many cases that there are not dense connections among the same labeled papers. The authors should check the correlation between the detected communities and the original paper labels.

2. No comparison with other community-preserving node embedding methods, such as "Community Preserving Network Embedding" in AAAI17

3. Since this paper aims to combine community detection and node embedding process, a set of baseline should be considered. For example, if considering the downstream node classification of node embedding as an evaluation task, then how about the performance of the following two-step method. We can first detect communities based on the node features then do graph node embedding by considering the communities' membership and node features together (e.g. simply concatenating both community membership features and node features).

4. Efficiency and scalability evaluations are needed. Spectral clustering has a scalability issue when meeting big graphs. Since the spectral process is also applied in the proposed method, efficiency and scalability evaluations are encouraged to provide, especially for big graphs which are not covered in the selected datasets in this paper.

5. In Sec 5.3 and Fig 2, it's mentioned that trends of the three datasets are different. For the increasing trend, how about the performance for an extreme case where all nodes are considered in one batch. On the other hand, adding more nodes in one minibatch could provide more information, but why there exists a decreasing trend? Though the authors provide a reason in Sec 5.3, it's better to analyze the reason directly from the datasets.

**Experience Assessment:**

I have published one or two papers in this area.

**Review Assessment: Checking Correctness Of Derivations And Theory:**

I assessed the sensibility of the derivations and theory.

**Review Assessment: Checking Correctness Of Experiments:**

I carefully checked the experiments.

**Review Assessment: Thoroughness In Paper Reading:**

I read the paper at least twice and used my best judgement in assessing the paper.

---

> ### Author Response · Authors · 2019-11-12
> **Response to Reviewer 3**
>
> We would like to thank the reviewer for these constructive comments.
> 1. The assumption that each class label is associated with a community is not entirely correct. First, there is no 1-to-1 mapping between a class label and a community. The 1-to-1 mapping relies on an assumption that the number of communities and the number of classes should be the same. There are cases that there are more communities than classes which makes n-to-1 mapping between communities and class labels. For instance, several small unconnected communities form a class label.
> Second, the node embeddings can be considered as unnormalized probabilities of assigning nodes to communities. Since the classifier is built on the node embeddings, there could be cases that all the nodes that have 60% chance to be in community C1 and 40% chance to be in community C2 to be classified to label L1 while another node with different probability assignments is assigned to another label.
> The assumption only makes sense when there are clear separation between communities in the graph and each community is associated with a label.
>
> 2. It is worth noting that the technique in [1] (which is called M-NMF) is an extended version of one baseline (NMF) that we have compared in the paper. However, we have added a comparison with the mentioned paper for completeness. Please see Table 2 in the updated paper. Still, our proposed method is able to outperform M-NMF significantly on all datasets.
>
> 3. In this paper, we do not aim to combine community detection and node embedding process. We aim to learn node embeddings by detecting community. In the baseline proposed by the reviewer, it requires concatenation between community membership and node features, which requires knowing the community structure before learning the node embeddings. Our experiment confirms that our approach outperforms the baseline. For instance, in the Cora dataset, DMC achieves 0.839 while this baseline's accuracy is 0.749.
>
> 4. It is worth noting that our proposed method (DMC) does not use spectral process. In our paper, we discuss spectral method as it provides an analytical solution to minimize the mincut loss. This means spectral method can serve as a baseline to compare our results obtained using neural networks with DMC.
>
> 5. It is expected that adding more nodes in a minibatch provide more information, which provides better performance. After analysing the dataset, we found out another reason for the decreasing trend for the Pubmed dataset that the node features are extremely useful for node classification. Adding more structure information (by adding nodes to the minibatch) would add noise, which decreases the classification accuracy. On the other hand, for other datasets, we obtain an increasing trend as expected.
>
>
> [1] Wang, Xiao, et al. "Community preserving network embedding." Thirty-First AAAI Conference on Artificial Intelligence. 2017.

---

### Official Review · AnonReviewer1 · 2019-10-23
**Official Blind Review #1**

**Rating:** 3

**Review:**

This work proposes a neural netowrk approach to minimize mincutloss, thus achieving embedding nodes and find communities at the same time. However, it is difficult for me to understand the paper and I feel that it is not clearly written.

1. The algorithm is not written in a box as in Algorithm 1. At first I thought algorithm 1 is the main method, but only after reading it I realized that it is one step of the algorithm. I would appreciate it if the complete algorithm (including input, output, parameters) can be summerized clearly.

2. I am confused about the claim "spectral approach underperforms significantly on the bibliographic datasets as it only uses the structure information in the graph." I thought the input of all methods are the adjencency matrix A.

3. In table 2 and table 4, why does the paper compare different methods with different measures? Is it possible to compare all methods using all measures?

**Experience Assessment:**

I do not know much about this area.

**Review Assessment: Checking Correctness Of Derivations And Theory:**

I did not assess the derivations or theory.

**Review Assessment: Checking Correctness Of Experiments:**

I assessed the sensibility of the experiments.

**Review Assessment: Thoroughness In Paper Reading:**

I read the paper at least twice and used my best judgement in assessing the paper.

---

> ### Author Response · Authors · 2019-11-12
> **Response to Reviewer 1**
>
> We agree that it would be hard to understand the paper if you are not from the field. We provide the summary of the paper as follows: In this paper, we propose a new unsupervised approach to learn node embeddings and their communities/clustering in an end-to-end manner. Traditionally, it would require 2 steps to obtain communities by first learn the node embeddings unsupervisingly and then apply a clustering technique. We achieve both objectives by learning an encoder in an end-to-end manner to minimize the mincut loss which aims for detecting communities. We also compare our approach with an analytical baseline using spectral method to minimize the mincut loss. We show that our node embeddings are better than embeddings learned by other baselines in several downstream tasks such as node classification. In addition, the detected communities are meaningful as shown in the word embeddings experiment.
>
> 1. We have also included the complete algorithm of our approach in the Appendix D. Please see the updated version.
>
> 2. There are two ways to encode nodes to their embeddings which are shallow and deep encoders. Shallow encoders are similar to word embeddings. Deep encoders such as Graph Neural Networks techniques require nodes to have features. This means in addition to the adjacency matrix of the graph, deep encoders also require the node features matrix as well. We have updated Section 3 to include this information for clarification.
>
> 3. Since our method (DMC) is able to achieve both node embeddings and node communities at the same time, we need to evaluate these two aspects in the experiment. In Table 2, different methods are compared in terms of the quality of node embeddings using node classification as the evaluation downstream task. In Table 4, we evaluate the second aspect which is the quality of the detected communities/clusters. In both measures, we show that our method outperforms other methods significantly.

---

### Public Comment · ~Filippo_Maria_Bianchi1 · 2019-11-08
**Some questions about the paper**

Dear authors,

I have read your paper since I am particularly interested on the topic and there are some questions I would like to ask.

1. The min cut loss has two types of degenerate solutions:
- the samples are uniformly assigned to all clusters
- some of the clusters are empty or even, all the samples are assigned to the same cluster.
The trick proposed to force the cluster assignment to resemble a one-hot vector accounts for the first problem, but I don't understand how the second degenerate solution is avoided in your framework.

2. In Theorem 2 it seems like there's an assumption that the adjacency matrix of the original graph and the adjacency matrix of a subgraph are binary and, most importantly, their elements are i.i.d. Is it correct?
An adjacency matrix (especially for an undirected graph) follows specific patterns. Furthermore, there is a strong dependency between the adjacency matrix of the original graph and the adjacency matrix of the subgraph.
Therefore, how reasonable are these assumptions in practice?

3. What are the differences between Theorem 1 in these paper and the theorems found in several spectral clustering papers (including those also cited by the authors), which state that the cluster assignments vectors that minimize the mincut objective correspond to the first eigenvectors of the Laplacian?

4. I didn't understand the difference between the node embeddings (E?) and the membership matrix (H). It's written that H is derived by a node embedding model and the embedding vectors have dimension d, however, the number of clusters should be k. I guess that the d-dimensional embeddings are those in E but, if that is the case, I can't understand how H is derived.

5. How sensitive is the temperature parameter \tau and how it should be selected in practice?

6. Could you explain to me what is the rationale behind using Gumbel-Softmax rather than just a Softmax? How the performance change in practice?

Thank you  :)

---

> ### Author Response · Authors · 2019-11-12
> **Thank you for your questions.**
>
> 1. Our approach does not suffer from both degenerate cases. The second degenerate solution is avoided by using the normalized cut (normcut). For the normcut, the case where all nodes are assigned to one community while the others are empty can not happen as this is not a valid solution. This is discussed in Section 3 - degenerated cases.
>
> 2. The i.i.d assumption for the graph is not new as it can be found in several works on random graphs such as the textbook [4]. It is reasonable as we do not have any information regarding the structure of the graph. In this case, whether there is a connection between two nodes can be considered as independent of other nodes. Since the original adjacency matrix is i.i.d, its submatrices are i.i.d as well.
>
> 3. We do not claim contribution on Theorem 1. Theorem 1 is provided for completeness as it discusses an analytical solution to the normcut loss. This spectral approach serves as a baseline which our neural network compares to.
>
> 4. The difference between the node embedding matrix E and the membership matrix H is their dimensionality. In many cases, the node embedding size d is usually large (e.g. 128, 256 or 512) as they correlate with the capacity of the model. On the other hand, the number of desired communities k is usually smaller than the node embedding size. As a result, to obtain the membership matrix from the node embedding matrix, we need at least a linear mapping between E and H. Other sophisticated mappings can also be used.
>
> 5. In our experiments, we observe that the temperature hyperparameter is very robust. This observation is similar to other works such as [1,2,3]. Following existing works, we set the temperature to be 1.0. We also experimented with temperature annealing but obtained no significant difference in performance.
>
> 6. There are two main reasons for the use of Gumbel-Softmax. First, it prevents the degenerated cases where all the nodes are assigned to all communities equally, which you have pointed out in the first comment. Second, the Gumbel-Softmax allows us to obtain clear community structure, which means we also obtain the community membership together with the node embeddings. This can not be achieved by traditional approach where another clustering step is required.
>
> [1] Choi, Jihun, Kang Min Yoo, and Sang-goo Lee. "Learning to compose task-specific tree structures." Thirty-Second AAAI Conference on Artificial Intelligence. 2018.
> [2] Gu, Jiatao, Daniel Jiwoong Im, and Victor OK Li. "Neural machine translation with gumbel-greedy decoding." Thirty-Second AAAI Conference on Artificial Intelligence. 2018.
> [3] Niu, Xing, Weijia Xu, and Marine Carpuat. "Bi-Directional Differentiable Input Reconstruction for Low-Resource Neural Machine Translation." Proceedings of the 2019 Conference of the North American Chapter of the Association for Computational Linguistics: Human Language Technologies, Volume 1 (Long and Short Papers). 2019.
> [4] Van Der Hofstad, Remco. Random graphs and complex networks. Vol. 1. Cambridge university press, 2016.

---

### Decision · Program_Chairs · 2019-12-19

**Decision:**

Reject

**Comment:**

The authors present an approach to learn node embeddings by minimising the mincut loss which ensures that the network simultaneously learns node representations and communities. To ensure scalability, the authors also propose an iterative process using mini-batches.

I think this is a good paper with interesting results.  However, I would suggest that the authors try to make it more accessible to a larger audience (2 reviewers have indicated that they had difficulty in following the paper). For example, while Theorem 1 and Theorem 2 are interesting they could have been completely pushed to the Appendix and it would have sufficed to say that your work/results are grounded in well-proven theorems as mentioned in 1 and 2.

I agree that the authors have done a good job of responding to reviewers' queries and addressed the main concerns. However, since the reviewers have unanimously given a low rating to this paper, I do not feel confident about overriding their rating and accepting this paper. Hence, at this point I will have to recommend that this paper cannot be accepted. This paper has good potential and the authors should submit it to another suitable venue soon.